# Collection of Controlled Nanosafety Data—The CoCoN-Database, a Tool to Assess Nanomaterial Hazard

**DOI:** 10.3390/nano12030441

**Published:** 2022-01-28

**Authors:** Harald F. Krug

**Affiliations:** NanoCASE GmbH, St. Gallerstr. 58, CH-9032 Engelburg, Switzerland; hfk@nanocase.ch; Tel.: +41-793565188

**Keywords:** nanomaterials, hazard assessment, database, lung toxicity, titanium dioxide, study quality

## Abstract

Hazard assessment is the first step in nanomaterial risk assessment. The overall number of studies on the biological effects of nanomaterials or innovative materials is steadily increasing and is above 40,000. Several databases have been established to make the amount of data manageable, but these are often highly specialized or can be used only by experts. This paper describes a new database which uses an already existing data collection of about 35,000 publications. The collection from the first phase between the years 2000 and 2013 contains about 11,000 articles and this number has been reduced by specific selection criteria. The resulting publications have been evaluated for their quality regarding the toxicological content and the experimental data have been extracted. In addition to material properties, the most important value to be extracted is the no-observed-adverse-effect-level (NOAEL) for in vivo and the no-observed-effect-concentration (NOEC) for in vitro studies. The correlation of the NOAEL/NOEC values with the nanomaterial properties and the investigated endpoints has been tested in projects such as the OECD-AOP project, where the available data for inflammatory responses have been analysed. In addition, special attention was paid to titanium dioxide particles and this example is used to show with searches for in vitro and in vivo experiments on possible lung toxicity what a typical result of a database query can look like. In this review, an emerging database is described that contains valuable information for nanomaterial hazard estimation and should aid in the progress of nanosafety research.

## 1. Introduction

The technical and chemical developments in nanotechnology have clearly demonstrated that the schedule from invention to market release has become much shorter compared to the last century. Additionally, the protection of the environment and human health against toxic chemicals or materials and regulatory demands have also been intensified. The combination of these two areas has conflict potential, thus, information about new technological developments is needed to establish a comprehensive risk assessment for new chemicals and innovative materials [1,2]. To support research and development activities in industry and science institutes, many funding programs support the accompanying risk research. In the case of nanotechnology, national and international action plans coordinate the funding of research on hazard and exposure of nanomaterials, e.g., the Nano Safety Cluster of the European Commission (https://www.nanosafetycluster.eu/, last access 8 December 2021). These activities have led to an enormous increase in published studies since the 2000s, as can be seen in Figure 1. In the PubMed literature database, there exists more than 40,000 published studies on a huge variety of nanomaterials investigated in different biological models, such as animals, cell and tissue cultures or plants. Keeping track of this tremendous number of published data is very difficult, both for scientists but even more for representatives of the industry or regulatory authorities [3]. In addition to the sheer number of publications, the quality issue is of great importance [4]. Not all published data are produced under good laboratory practice, or even under good scientific practice conditions. Many labs do not use standard operating procedures (SOPs) or harmonized protocols for toxicological studies, although many projects funded in Europe for example produced such standardized methods and published the protocols on different websites (e.g., the national DaNa project https://nanopartikel.info/en/knowledge/operating-instructions accessed on 8 December 2021 and the European PATROLS-GRACIOUS project consortium—https://www.nanosafetycluster.eu/joint-patrols-gracious-nanosafety-cluster-event-on-harmonization-of-standard-operating-procedures-sops/ accessed on 8 December 2021).

The constant growth of data and the diversity in quality requires a way to use the existing knowledge effectively and properly. Therefore, several projects and activities in recent years have established many databases on nanomaterials with different contents (Table 1). It has already been discussed in detail that it is not easy to decide which database fits best to the needs of a specific user [5]. Besides the fact that most databases contain information about products containing nanomaterials or show relationships between nanomaterial properties and possible applications, good toxicological data are often missing. Only three of the databases available online contain toxicological data, these are eNanoMapper, NanoE-Tox and NBIK. Whereas NBIK is exclusively contains data from zebrafish embryo exposure experiments [6], NanoE-Tox contains ecotoxicological data on a huge variety of animal and plant species [7]. eNanoMapper delivers further additional toxicological relevant data an many species and information about ontology and material characteristics [8]. The two databases on ecotoxicological data are very special on the one hand but, on the other hand will no longer be updated. The only living database with toxicological relevant data is eNanoMapper but it is relatively complex and can mainly only be used by experts. Furthermore, there are no comparative values as result of a query, that could indicate whether a nanomaterial poses a hazard or not.

All these reasons have forced the consideration of the establishment of a new database, as the relationship between the material properties and the biological effects that may be induced can be very informative. To make scientific results especially available for users outside the scientific community without excluding scientists, the CoCoN^®^ database is to be created based on the “Collection of Controlled Nanosafety Data”. Searching for toxicological data related to the properties of specific nanomaterials should be possible and the result would ideally have to be a comparable value that has good validity for different biological models. One such value could be the NOAEL in animal studies and the NOEC in cell or tissue culture experiments [9]. During the last 8 years, a very large data collection has been developing, containing several thousands of datasets on many different nanomaterials, tested in various biological models for their toxicological potential. This data collection is the basis for the programming of the CoCoN^®^ database, which aims to deliver a material property-related outcome. The CoCoN^®^ database, which is an active development, will deliver quality-assessed data on a huge variety of biological endpoints affected by nanomaterials which will be taken from the available published literature. This paper describes the idea behind CoCoN^®^ and provides use case examples. The database should help to develop a better understanding of the material property to hazard relationship and its query tools may produce new knowledge for predicting the toxic potential of advanced materials.

## 2. Materials and Methods

The data collection which will be used for the establishing of the database has been generated in several phases. The first phase reflects the literature between 2000 and 2013. During this phase the focus was on all materials without exception and some of the results have already been published [10]. In the year 2019 in a second phase the search was extended for the years 2014 to 2018. From the literature found in the second phase, only material specific evaluation has been carried out until now. Thus, the data presented here mainly concern the first phase if not mentioned otherwise.

### 2.1. Literature Search and Study Selection

In the first phase, the literature was taken only from the PubMed database and imported into an Endnote Library. The search profile is shown in Table 2. The number of found citations with this search profile per year can be seen in Figure 1. For the first phase it sums up to 11,058 and for the second phase to 24,303 publications.

As mentioned above in the first phase all investigated nanomaterials have been recorded. The number of studies was reduced by a defined selection process focused on toxicological investigations on nanomaterials. The exclusion criteria were defined as follows:The study deals with medical treatment or therapeutic application of the nanomaterial only, e.g., drug delivery or therapeutic application via injection.No real toxicological experiments have been shown; this is the case if a material- or engineering-oriented publication uses the buzz word “toxicity” or similar within the introduction or discussion, which led to the inclusion of the publication at the beginning.Neither animal studies nor cell- or tissue culture experiments have been described, which excludes reviews as well.The content deals exclusively with environmental issues, e.g., air pollution with diesel exhaust particles or ultrafine dust, or describes solely ecotoxicological topics such as plant research or distribution in environmental compartments such as water, soil or air.The publication is not available as pdf file and/or not written in English or German language.

Based on this selection process the number of studies was reduced from 11,058 to 6626 for the first project phase [11]. To better handle the huge number of citations the excluded non-relevant publications have been removed from the library.

In the second phase more than 24,000 publications have been found with the same search profile and included into a second Endnote library. The number of publications of the second phase has not been reduced with the same selection process as described above. Until now only a rough selection (e.g., wrong language, etc.) has been carried out and the number of citations in this library is actually 19,732. These newer studies are the basis for specific searches on defined materials such as silica, graphene, or titanium dioxide. As the second library is not at the same stage as the first one, the following examples and results concentrate only on publications from the first phase but represent examples for the CoCoN^®^ database as it is planned to be established.

### 2.2. Data Extraction and Collection

The publications from the first phase have been implemented into a huge table with four different sections. The first section contains bibliographic data such as author names, publication year, and the digital object identifier (DOI) as a specific identifier. Moreover, information about the study type (in vitro or in vivo), the intended application of the nanomaterial (food sector, cosmetics, technical applications, etc.), and a first indication of whether this study is toxicology oriented or more of a mechanistic study is collected here. The second part presents all the available information about the used material. Name, size of primary particles, source, shape, surface charge and specific surface area, crystal phase, coatings, aggregation or agglomerations state, the used concentrations within the experiments and some more information are given in this section. Within the third part, the information about the biological model is presented. Animal species for in vivo studies or cell type for in vitro studies, the type of culture, the treatment design with information on duration, repeats, exposure pathways, dose concentrations and OECD guideline number, if available, and more. All these data are included as the authors have mentioned them in the publication. Following this fundamental information about the respective study, the biological endpoints are then recorded with their associated assays.

Each publication may result in several datasets within this table because each investigated material examined, and even each variation of it, e.g., different sizes or different coatings, are recorded as a separate dataset. It may happen that one publication generates a huge number of datasets when many materials have been tested in a large number of cell cultures (e.g., [12]; with more than 200 datasets) but on average there are 2 to 4 datasets per publication.

### 2.3. Quality Evaluation—Scoring

During the collection of the data, (readout) the quality of the study was evaluated. During the first phase of the project, a rough estimation was made about the quality just by looking for the obligatory characterization of the nanomaterial and the minimum information about the biological model. Later, after several national and international projects established criteria catalogues for the data quality of published studies (literature criteria catalogue from the German DaNa-project: https://nanopartikel.info/en/knowledge/literature-criteria-checklist/ accessed on 8 December 2021 [13]; the European GuideNano project established an Excel-Sheet for the quantitative evaluation of studies on nanomaterials [14]; the ToxRTool, established by a European project to refine the Klimisch scores for toxicological studies [15]). The studies were also evaluated for quality when the datasets were implemented into the existing table, which led to a disappointing outcome, in that more than 70% of the toxicological and mechanistical studies on nanomaterials were not reliable [16]. Taken together, the most important criteria from the different criteria catalogues and tools for the CoCoN^®^ database a variation of the GuideNano-evaluation scheme have been used to automatically generate a quality score which is attached to the publication for the traceability of the process. The quality evaluation was performed in four different sections and for each tox category (in vivo, in vitro, and ecotox) there was a value calculated separately in combination with the material characterization. The rating can take the following values (Figure 2), where 0 and 0.5 are low quality studies, 0.8 are good studies and 1 is the value for very reliable studies.

The score is an important value which is included in the table (and later in the established database as well). This gives the user of such a database the chance to select “all” available data or “only the good” studies for their queries.

### 2.4. NOAEL/NOEC—Readout from Published Studies

The heart of the database and thus also the most important result is the possibility to readout a real comparative value between the studies. For toxicological assessments, the dose or concentration at which a biological effect is just not triggered is of importance. This no-observed adverse-effect-level (NOAEL—in vivo) or no-observed-effect-concentration (NOEC—in vitro) is included from each study for the observed biological effect or key event which is described by the authors. One difficulty for the readout of this value from published data is the fact that not all studies cover the concentration or dose range to directly determine the NOAEL/NOEC. As shown in Figure 3, there exist mainly three variations. In the best case, the concentration range is chosen correctly by the authors and the NOAEL/NOEC can be read off from the data directly (Figure 3a). The second case is that an effect was observed even at the lowest concentration used in the experiment. In this situation, the NOAEL/NOEC is lower than the lowest concentration, which in this case represents the lowest-observed-adverse-effect-level (LOAEL) or lowest-observed-effect-concentration (LOEC). In such a case the value in the table is given with “<” to demonstrate that there might be a concentration less than the number extracted from the publication (Figure 3b). The last case consists of the fact that none of the used concentrations changes the studied biological activity, that is, not even the highest one. Thus, the highest concentration used is the upper-no-observed-adverse-effect-level (UNOAEL) or upper-no-observed-effect-concentration (UNOEC), which means that there is no toxic effect at all (Figure 3c). To show this in the table, the respective value is preceded by the character “>”.

During the extraction of the data from the publications, a major hurdle to the usability of the data became apparent, namely the information on the dose or concentration units used. These are generally not standardized in the studies and therefore differ greatly Table 3). Moreover, the exposure pathway is different too, especially for the exposure pathway via the air, as there exists a huge variety of application methods. In addition to inhalation (nose-only or whole-body exposure) and instillation (intranasal, intratracheal, or oropharyngeal), the authors of the studies have used aspiration (pharyngeal or oropharyngeal), and intrapulmonary spraying. Wherever possible, units have been recalculated to finally result in a greater number of matching dose units.

## 3. Results

Currently, more than 1500 studies of the first phase have been evaluated and included in the data collection, which represent more than 4000 datasets. Additionally, more than 2000 studies from the second phase have been evaluated for specific materials and 888 have been included in the data collection, representing 1318 datasets for these specific nanomaterials chosen. During the next phase of programming the new CoCoN^®^ database, more studies will be included taken from the pre-evaluated selection of both phases. In the following, only evaluations of studies from the first phase are presented, as they were evaluated with the same approach and provide a consistent picture.

### 3.1. Key Figures of the Data Collection

The more than 4000 datasets of the first phase regarding the literature between 2000 and 2013 show results for more than 100 different nanomaterials which have been investigated. To really put this number in perspective, it is important to know that for this calculation, e.g., all multi-walled carbon nanotubes are taken as “One” material. The same applies for graphene, titanium dioxide, silica, and all other materials. Discriminating between different crystallinities, fiber lengths, oxidation states and other properties of the nanomaterials would result in a much larger number of materials. Additionally, more than 150 different coatings have been found, but most of them have been used only once or less than five times. A comparable high number of cell cultures have been used for the in vitro studies, as around 260 different cell lines are treated with nanomaterials. The variability is much less for animal studies, where mainly rats, mice, fishes, nematodes, daphnia, mussels, or drosophila have been investigated. The biological endpoints, which are described within the studies, sum up currently to 107 in the table with the results, but are still increasing. For the future database, some of the endpoints will have more than one choice concerning gene expression, for example, other omics technologies which result in several alterations of genes or proteins, which are probably up or down regulated. At the end of the table a selection can be made during input of data which pathways of toxicity may be involved and which adverse outcome pathway could be the result of the biological effects in the investigated system. The input is mainly based on the statements of the authors. If there is no clear statement or another possibility to derive this information from the data, then these fields are left blank.

Not all the information provided by the authors is unambiguous. The most obvious indication with high potential for misunderstandings is to present the applied amount of nanomaterial to cell cultures in molarities. It is very difficult to re-calculate the given concentration to µg/mL or similar for TiO_2_ nanoparticles [17,18] but the question remains as to how to re-calculate the molarity of multi-walled carbon nanotubes into a comparable dimension [19]. Such concentration units are not meaningful and are irrelevant to hazard or risk assessment. Perhaps these papers should be rejected during the review process because the most important part of the experiments is not comprehensible. In addition, the concentration data in µg/mL for in vitro experiments are not always unambiguous and traceable. Without the knowledge of which Petri dishes or multi-well plates have been used, how much volume has been applied per dish, and how many cells have been plated into these dishes, the exact design of the experiment cannot be reproduced. Without the above-mentioned details, the real concentrations the cells are treated with may vary by more than 100%. Such details become obvious when the studies are re-evaluated and analyzed during data extraction for the database.

### 3.2. Readout Results from the Data Collection

A first look at the distribution of the studies in relation to the endpoints gives a clear picture of the focus of the studies. Specifically, the in vitro studies have clear priorities in the effects studied (Figure 4). The overall data collection has been the basis for several projects and activities in recent years. Therefore, an excerpt will be given here concerning which possibilities exist to be able to use the data of the collection.

#### 3.2.1. OECD-Project on Adverse Outcome Pathways for Nanomaterial Risk Assessment

Besides the already published data [10,16,20], one of the first intense queries has been made for the international OECD WPMN Pilot Project on Advancing Adverse Outcome Pathway (AOP) Development for Nanomaterial Risk Assessment and Categorization [11]. To follow the selection process for such specific queries, the PRISMA system has been applied [21]. From the 6626 Studies at the end, 191 were selected for the adverse outcome “inflammation”, which resulted in 447 datasets. The main goal of this activity was developing criteria and methods to identify and prioritize potential key events (KEs) from the nanotoxicity literature, which may be inflammation-associated. Within this subset of data around 60 different endpoints have been reported for 45 different nanomaterials and the KEs have been prioritized for their contribution to the inflammation process [11]. Additionally, for a selection of seven nanomaterials specific criteria have been discussed. Which endpoints are relevant for this AOP and if these are induced or not have been some of the key questions. The results on these questions have been published in 2020 [22]. Moreover, the relationship between the literature data extracted from the data collection and the possibility to assess the nanomaterials toxicity via non-animal strategies is another idea to use these datasets [23]. There exists another example in using data from dossiers to assess possible toxic activities of chemicals with the tools of machine learning and data mining. The group of Thomas Hartung at the Johns Hopkins University in Baltimore evaluated ECHA-dossiers, extracted the toxicity data, and calculated based on structure similarities the possible toxic potential of new chemicals and outperformed by this process the results of animal testing [24].

#### 3.2.2. Evaluation of Titanium Dioxide Lung Toxicity

Another example at this point concerns the evaluation of data on titanium dioxide. This material, which has been on the market for more than five decades, has repeatedly been the subject of discussion, especially since it has also been used as a nanomaterial directly in cosmetics and food. To specifically compare in vitro and in vivo results in this chapter, some possibilities will be demonstrated concerning how the display of the results of a query with appropriate filters could be presented.

The filter for the in vitro data was as follows:Material = titanium dioxide;Type of study = in vitro studies;Cell type = mammalian lung cells (exclusively macrophages or immune cells);Concentration units = µg/mL (or µg/cm^2^, if recalculation is possible);Biological endpoints = acute cytotoxicity, formation of reactive oxygen species (ROS), cytokine production;

The filter for the in vivo data was as follows:Material = titanium dioxide;Type of study = in vivo studies;Species = rats or mice;Exposure = inhalation (mg/m^3^) or;Exposure = instillation or aspiration (µg/kg or µg/animal);Biological endpoint = immune cell migration (neutrophil or macrophage influx).

The query for the in vitro studies revealed 56, 51, and 32 datasets for the three different endpoints: cytotoxicity, ROS formation and cytokine production, respectively. The database offers the possibility to distinguish between different size groups of the investigated nanomaterials. In this case, the primary particle sizes of the applied titanium dioxide variants were assigned to the following five size groups:0–10 nm11–20 nm21–50 nm51–100 nm101–500 nm

The data are shown in Figure 5 and demonstrate a typical result for a corresponding query from the data collection. Although the number of data points for cytokine production is relatively low, all three endpoints tell the same story: in most cases, an effect is triggered only at very high doses. This is particularly well illustrated if the work of Klaus Wittmaack is taken into account [25,26], who was able to show impressively that at concentrations higher than 27 µg/mL of titanium dioxide the cells in the Petri dish are “buried” under a layer of sediment particles, which prevents the normal supply of nutrients and oxygen to the cells. This is what the author calls the “landslide effect”, as the cells under such a layer of solid material will show signs of deficiency and start to die after some period of treatment. Nevertheless, most cell lines are robust and survive for the first 24 h, as can be seen by most data points beyond this limit shown in Figure 5A–C. Therefore, the statement from the in vitro studies is that titanium dioxide is not critical to lung cells and belongs to the nanomaterials of low toxicity.

The query for the in vivo studies revealed 25 datasets, 17 for instillation or aspiration and 8 for inhalation experiments from 14 studies in total. Ten Studies have been published in this period between 2000 and 2013 on the exposure pathway instillation or aspiration. The applied doses spread over a wide range from 400 to 5000 µg/kg for rats and 100 to 30,000 µg/kg for mice. The method of exposure via instillation or aspiration is paralleled by a very fast dose rate [27] and thus, it can be expected that the material provokes in most of the treatments a transient inflammatory response because of the delivery of a high amount of particles in a very short time. This usually manifests in immune cell migration into the lung and cytokine production by these cells. Two studies which have been carried out with mice used overload concentrations (≥500 µg/mouse lung), whereas no overload conditions have been applied in the experiments with rats. The results in detail can be looked up in the Appendix A), but here the most important facts will be listed:The overall number of studies with a comparable experimental study design is low; most toxicological studies do not yet use standardised protocols, e.g., OECD testing guidelines as suggested recently [28].The variety of investigated titanium dioxide is high; in this period, not even two studies used the same material.In rats, NOAEL range from 200 µg/kg to 5000 µg/kg, which might represent the huge variety in the materials.Some studies found a response for the only investigated dose; in many cases the observation time was only 1 d and in this treatment period the influx of neutrophils can be expected after exposure with dust particles as a “normal defence response”.For mice, the NOAEL is between 30 µg/animal to 500 µg/animal, which represents the overload dose.In mice and rats, the inflammatory response is strongly dependent on the material tested.For the more realistic inhalation studies, 7 datasets out of 8 do not describe any cell migration into the lungs; the only study which found an inflammatory response used 100 mg/m^3^, 6 h per day for 5 days for the inhalation exposure.

There are some more animal studies in the first phase of this data collection regarding lung exposure with TiO_2_, but these described other endpoints or used slightly different material shapes (e.g., rods or short fibres). A total of 29 publications deal with instillation/aspiration experiments and 26 publications are on inhalation experiments (reference list is given in the Appendix A). Nevertheless, it is very interesting to evaluate these data and consider the authors’ assessment when it comes to the overall impression of the effect of TiO_2_. Based on the described effects, a categorization was attempted to evaluate the effects in terms of an adverse outcome. The following five categories were defined, and the results are shown in Figure 6:Overload: this reflects the situation that for mice a higher dose than 500 µg/lung and for rats more than 2500 µg/lung were applied.No effect: describes the situation that even the highest dose used for the experiments did not induced any adverse effect in the animals during the observation time.GBP-like: the behaviour as a granular biopersistent particle (GBP) is described to induce a transient inflammatory response, which usually subsides after about 7 days and the animal recovers completely without permanent sequelae.Quartz-like: the relationship to quartz describes a more intense inflammation accompanied by oxidative stress, a formation of granuloma with signs of fibrosis.Asbestos-like: the effects which indicate quartz-like mechanisms and additionally genotoxicity and the induction of a higher tumour incidence.

The use of the data collection enables clear presentations of the results as shown above. The example of titanium dioxide and its effects to lungs and lung cells makes it clear that the relatively large number of data sets allows a good statement to be made about the effects of a specific material. Significant is the value for the NOAEL/NOEC which allows considerations for the classification in the Globally Harmonised System for Hazard Classification. A complete list of the references for all studies, for the in vitro as well as the in vivo studies, are given within the Appendix A.

## 4. Discussion

The tremendous development in the number of publications overall, but also for nanotoxicological topics, gives rise to considerations on how this flood of data could be harnessed. Moreover, the ongoing discussion on the safety of nanomaterials and innovative materials is often driven by single investigative studies with questionable toxicological backgrounds [29]. Based on these facts, there exists several activities in establishing databases for a variety of applications and different user groups (compare Table 1). Keeping in mind the registration of new products and materials during the REACH process (Registration, Evaluation, Authorisation and Restriction of Chemicals) it is of utmost importance that industry on the one hand and the registration authorities on the other have access to already prepared data [3]. This was the reason to establish the data collection under the aspect that a quick access can take place and important information can be queried in simple representation. So far, the existing tables containing most of the datasets is much too complex for such queries by clients themselves. Thus, the idea was born to transfer the datasets to a database in order to fulfil the criteria of easy access for clients and present an informative representation of the results.

In order to keep the complexity of the data in the database in check, certain details of the studies are not recorded (e.g., the in-depth details of the methods used in the studies), but the emphasis is on the information which is of great importance (e.g., the test design such as treatment of animals, duration, repeats, etc.). The scheme of the data collection follows the already published workflow for “nanodata” curation (Figure 7) as described a few years ago [30]. The existing data collection cannot and does not claim to be complete but gives an impression of the existing knowledge in its entirety. As soon as the database has been established and data entry has been simplified, the information from the clients should also be taken up (feedback system) when it comes to updating the content of the database and incorporating new studies.

For the incorporation of new study results into CoCoN^®^ in the future, it is of utmost importance that toxicological publications respect the FAIR principles. This would make the data needed for the database input findable, accessible, interoperable, and reusable [31]. The introduction of electronic lab books and accompanying Appendix A with publications in the form of toxicological dossiers with the most important information about the material, the biological model and test design would make it additionally much easier to implement such data into databases [32,33]. It was impressively demonstrated that the data from machine-readable REACH dossiers with the aid of machine learning processes provided a better estimate of the toxicological potential of chemicals than the corresponding animal tests [24]. This example illustrates the opportunity that lies in digitizing existing data. Of course, this will not replace all cell culture or animal experiments, but the evaluation and calculation based on the existing studies will help to significantly advance read-across [34,35] and other possibilities for risk assessment, following the 3R principles [36]. However, some efforts are still needed to achieve this goal, particularly as the range of variation in the studies on nanotoxicology published to date is very large. Test models, test design, materials used, and the exposure conditions mean that hardly any study is comparable to another, making it difficult to combine the results. In addition, studies often described as toxicological studies do not follow the principles of toxicology as demanded a long time ago [37]. This includes missing concentration dependencies or dose–response relationships, as well as the complete lack of standardized procedures [28]. Thus, mechanistic studies are very often overlooked in the literature as toxicological ones, which leads to a misjudgement from a toxicological point of view. This explains why the assessment of a substance could be understood based exclusively on high-dose experiments or, in the case of nanomaterials, so-called “overload conditions”.

In addition to these very conspicuous errors, however, there are other, less easily recognizable flaws that lead to the disqualification of published studies [38]. Very often, for example, the analysis of the properties of the nanomaterials used for the investigations is weak, right down to the size information, which can also be missing. However, without a reference of the results to the properties of the material, a study is of little value from a toxicologist point of view. Therefore, it is of importance to also look at and evaluate the quality of the studies when collecting data for a data base.

There exist many suggestions on how to increase the quality of studies, and the most important prerequisites for good quality papers on the toxicology of nanomaterials are listed here and have been published elsewhere [4,14,39,40,41]:A rigorous and adequate physicochemical characterization of the test materials is needed;Adequate particle controls must be included;Possible contaminations, such as endotoxins, should be analysed;Interferences of the tested material with the assay should be investigated;High-dose experiments designed to produce toxicological effects—which are publishable (and sensational)—should be avoidedAs far as possible, standardized protocols should be used to better compare results

These and more recommendations are also called for by project consortia and experts in the field. To face the problems of costly development of products and the pace regulators are confronted with, new approach methodologies are a possible solution for the improvement of risk assessment [1,2,3,42]. Additionally, here the implementation of searchable databases plays a central role. Such databases, as the herewith introduced CoCoN^®^ database, can provide answers to classify the biological effect of a material, but also allow read-across or grouping. As shown above for the example material titanium dioxide, the data can be filtered for a specific material, specific cell types and special biological endpoints. Figure 5 demonstrates clearly that the majority of datapoints show low toxicity and the mean value of the individual size classes is well above the maximum reasonable concentration of 27 µg/mL, as given by Wittmaack [25].

The presented results of the selected in vivo studies from 2000 up to 2013 (Appendix A) show typical characteristics for lung exposure experiments. First, the study design of instillation or aspiration induced a high dose rate, which usually induces an inflammatory response to particle exposure, which decreases during the first week after treatment [27]. This response is not observed within inhalation experiments that reflect more realistic dose rates, except for only one dataset (#1, Appendix A). Here, an extremely high dose of 100 mg/m^3^ for 6 h per day and 5 consecutive exposure days was used for this study. Secondly, many of the investigated materials are self-made materials or materials not allowed to be used on the market. Thirdly, the collection of in vivo data also shows the weaknesses of some studies which even do not announce the doses applied for the experiments shown in the figures of the publication (datasets 10 and 11, Appendix A, taken from [43]). 

Nevertheless, the following period from 2013 to now is of importance as well, and these studies are under evaluation. It should soon be possible to compare the data from the first phase with those from more actual studies and to draw further conclusions on the biological effects of titanium dioxide and other materials.

## 5. Conclusions

Safety concerns in the use of nanomaterials must be considered and their potential risks to human health as well as their undesirable effects on the environment should be avoided. To assess such risks, the totality of published data is a good source, but a compilation of all existing data is lacking. Therefore, the CoCoN^®^ database is an attempt to process the already existing but very complex data collection in such a way that an easy and user-friendly retrieval of the study results is possible. This allows users of the database to enter their own queries and search for results for specific materials. This will allow the assessment of a potential hazard for nanomaterials or innovative materials and make the regulatory processes smoother and faster. Moreover, producers of new materials may get a first picture of the biological effects of their material and can assess how useful the further development of new material will be for specific applications. In the best case, the use of the metadata from the database can enable a cross-comparison of important details, both in terms of material properties and hazard potentials [44]. Scientists can use the data to make comparisons with their own work or between different materials, which may support grouping and read-across. Regulators can draw conclusions about material safety from their queries and manufacturers can identify and close existing knowledge gaps at an early stage of product development.

## Figures and Tables

**Figure 1 nanomaterials-12-00441-f001:**
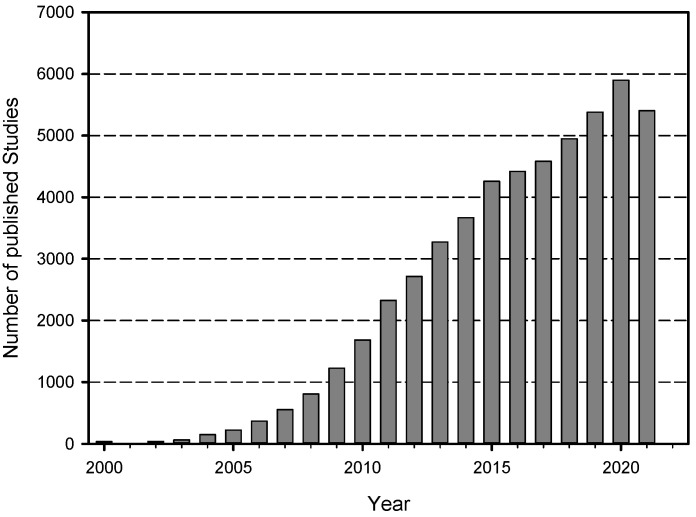
Publications per year found with a specific search profile in PubMed for the topic “Nanotoxicology”. The bar for the year 2021 represents all publications until the 4th of November of this year.

**Figure 2 nanomaterials-12-00441-f002:**
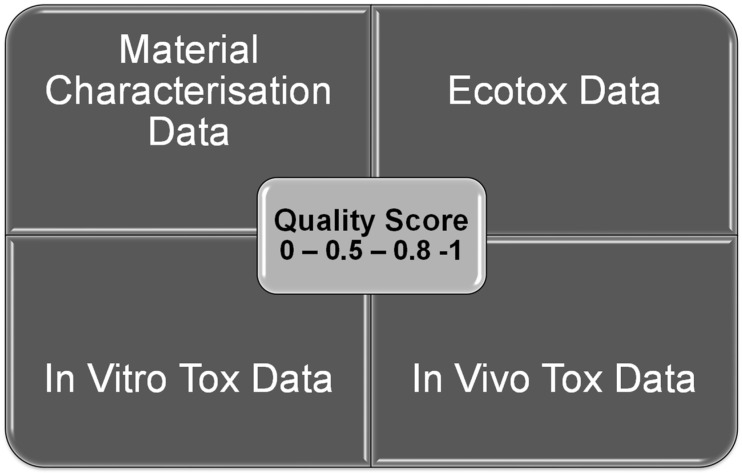
A scheme of the four sections for which quality scores are generated. Each tox-section calculated together with the material characterisation reveals a quality score for the specific kind of study.

**Figure 3 nanomaterials-12-00441-f003:**
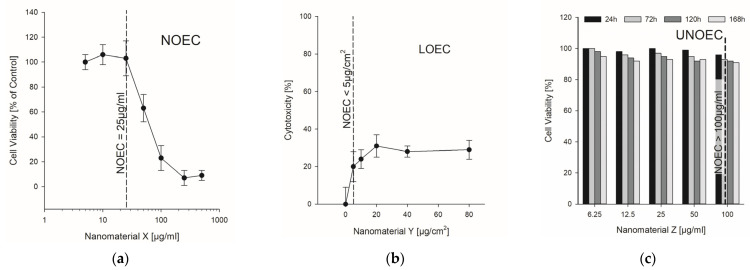
An explanation of how the NOAEL/NOEC were readout from the data presented within the published studies. The three examples shown here are based on real data taken from publications but reproduced without any relation to the materials and the publications, as these are just to demonstrate the process. All three examples are in vitro studies and measured cell viability or cytotoxicity. On the left (**a**) the NOEC can be determined directly from the graph as 25 µg/mL. In the middle (**b**) the lowest concentration used is still affecting the cells, thus it represents the LOEC. On the right (**c**) even the highest concentration does not induce any toxic effect on the cells and the value of 100 µg/mL is the UNOEC.

**Figure 4 nanomaterials-12-00441-f004:**
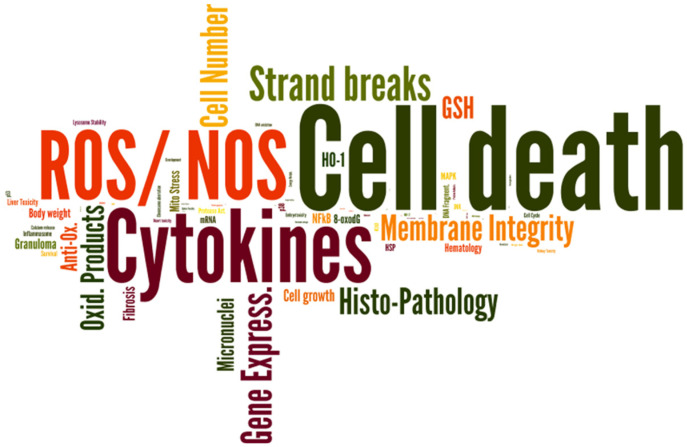
Word Cloud results after computing of all in vitro experiments within the data collection. Letter sizes are directly proportional to the number of data sets for the respective analyzed endpoint.

**Figure 5 nanomaterials-12-00441-f005:**
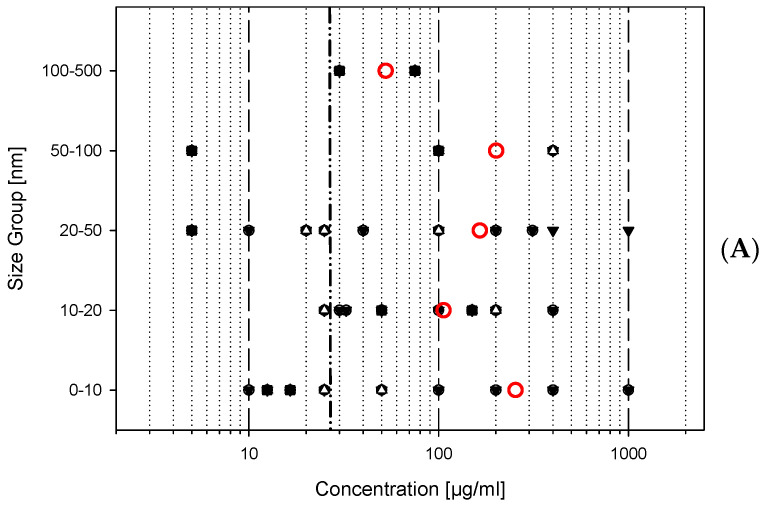
NOEC for titanium dioxide treatment of mammalian lung cells. Shown are 56 datapoints for cytotoxicity (**A**), 51 datapoints for ROS formation (**B**) and 32 for cytokine production as extracted from the data collection with the specific filter given in the text. Abscissa gives the concentrations at which no effect could be observed in µg/mL (NOEC). The ordinate shows the result for five (**A**,**B**) or four different size groups (**C**) of the primary particle sizes as given by the authors. Often, multiple data points overlie each other. The red open circles represent the calculated mean for each size class. No data available for the size class 100–500 (**C**) The dash-dotted-vertical line (— ●● —) represents the limit for the “landslide” situation above 27 µg/mL (explanation see text).

**Figure 6 nanomaterials-12-00441-f006:**
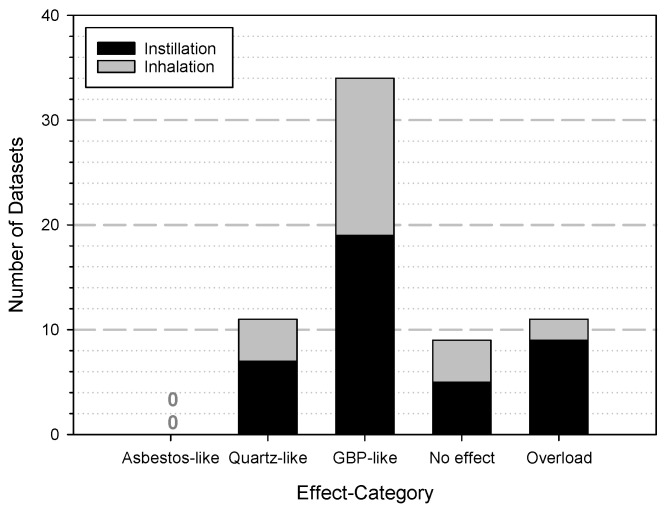
The attempt to categorize the effects of TiO_2_ in lung exposure studies. The distribution of the datasets from the data collection over the five different categories is shown. Black bars show instillation experiments and grey bars the data for inhalation studies. For definition of the categories see text.

**Figure 7 nanomaterials-12-00441-f007:**
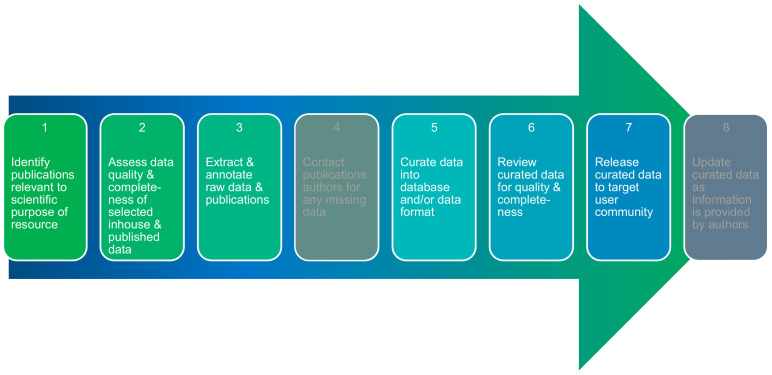
The workflow during data collection and implementation into the CoCoN^®^ database. Steps 4 and 8 in this process were cancelled as it takes too much time to contact the authors to complete their data for so many publications. Figure 7 was adapted from [30] (“Nanocuration workflows: Establishing best practices for identifying, inputting, and sharing data to inform decisions on nanomaterials”, © 2015 C. M. Powers et al., distributed under the terms of the Creative Commons Attribution 2.0 International License, https://creativecommons.org/licenses/by/2.0).

**Table 1 nanomaterials-12-00441-t001:** Popular databases of nanomaterials (all the web links were accessed on 8 December 2021).

Databases	Website	Remark
caNanoLab	https://cananolab.nci.nih.gov/	Nanotechnology in biomedical research on cancer
eNanoMapper	https://data.enanomapper.net/	Ontology and safety assessment of nanomaterials
NR	https://nanomaterialregistry.net/	Physicochemical properties of selected nanomaterials
NanoData	https://nanodata.echa.europa.eu/	Nanotechnology in products in 8 different sectors
Nanodatabase	https://nanodb.dk/en/	Nanomaterials in products
NanoE-Tox	http://www.beilstein-journals.org/bjnano/content/supplementary/2190-4286-6-183-S2.xls	Ecotoxicological data available as an excel sheet which will not be updated anymore
NanoNature	https://nano.nature.com/	Literature database on nanomaterials; will be retired in June 2022
Nanowatch	https://www.bund.net/themen/chemie/nanotechnologie/nanoprodukte-im-alltag/nanoproduktdatenbank/	Commercially available products containing nanomaterials; available in German only
Nanowerk	https://www.nanowerk.com/	Database on suppliers of commercially available nanomaterials
NBIK	http://nbi.oregonstate.edu/ no safe connection on access day	Database on study results of nanomaterial exposure effects in embryo zebrafish
NECID	https://necid.ifa.dguv.de/Login.aspx?ReturnUrl=%2fUser%2fFirstpage.aspx	Data on occupational nanomaterial exposure during various exposure scenarios
NIL	http://nanoparticlelibrary.net/	Physicochemical characteristics of very specifically produced nanomaterials
NKB	https://ssl.biomax.de/nanocommons/	Nano-safety knowledge infrastructure
PubVINAS	http://www.pubvinas.com/ no safe connection on access day	Virtual nanostructure simulation tool
PaFTox	https://publica.fraunhofer.de/dokumente/N-277711.html	Data on genotoxicity of nanomaterials; the database is not available anymore although funded by government money
StatNano	https://statnano.com/	Applications and properties of nanomaterials

**Table 2 nanomaterials-12-00441-t002:** Search profile in Endnote for PubMed database for publications on nanotoxicological studies.

Field Delimiter	Where	How	What
	All Fields	Contains	nanotox *
Or	All Fields	Contains	fulleren * AND toxic *
Or	All Fields	Contains	carbo nanotube * AND toxic *
Or	All Fields	Contains	bucky ball * AND toxic
Or	All Fields	Contains	nanotube * AND toxic
Or	All Fields	Contains	nanoparticle * AND toxic *
Or	All Fields	Contains	nanomat * AND toxic *
Or	All Fields	Contains	Nano * AND toxic
Or	Year	Contains	2021 ^1^

^1^ for 2021 5515 records have been found on the 11th of November. *: wildcard for the search words.

**Table 3 nanomaterials-12-00441-t003:** Quantities in the dosing of nanomaterials in the in vitro and in vivo experiments as given by the authors.

Given Concentration Units for In Vitro Studies	Given Dose Units for In Vivo Studies	
µg/mL	oral exposure	inhalation/instillation
µg/cm^2^	µg/animal	µg/m^3^
cm^2^/mL	µg/kg BW and	mg/m^3^
m^2^/cm^2^	mg/kg BW	cm^2^/m^3^
mM/µM/nM	dermal or intradermal	/cm^3^
ppm	µg/µL or mL	µg/animal
/mL	g or mg/ear	µg/g lung tissue
/cm^2^	mg/animal	µg/lung
/cell	mg/kg	

## Data Availability

The data points in all figures have been taken directly from original publications which are included in the CoCoN data collection. The selected contributions used for this study are listed in the Appendix A. Further inquiries can be directed to the corresponding author.

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
