# Peer review of "Collection of Controlled Nanosafety Data—The CoCoN-Database, a Tool to Assess Nanomaterial Hazard"

_nanomaterials, 2022, doi:10.3390/nano12030441_

Round 1
Reviewer 1 Report
My main concerns are:
- Part of the text in Figure 3 is blocked, and a “h” is missing after 120, please modify it carefully.
- The abstract should differ from the conclusion in language expression.
- The keywords and title need to be revised.
- The language of the manuscript should be concise.
- Increase the number of references appropriately.
- Some references need to be updated.
Author Response
- Part of the text in Figure 3 is blocked, and a “h” is missing after 120, please modify it carefully.
I do not fully understand, what the reviewer has meant with the “blocked text” in Figure 3. All the text within the figure is blocked, so no changes can be made. The missing “h” I have added in the new version. - The abstract should differ from the conclusion in language expression.
Since both reviewers raised the same issue for the abstract, I have written a new one following the suggestions of the reviewer. - The keywords and title need to be revised.
I omitted “nanotoxicology” from the keywords as suggested and have added to the title - The language of the manuscript should be concise.
I have thoroughly revised the text and changed all positions where it was necessary. See point by point list below - Increase the number of references appropriately.
I have added 11 new citations at important positions (Nos. 1,2,3,9,13,28,32,33,34,35,37 in the new reference list) and another I have cited earlier (3) - Some references need to be updated.
Usually I try to cite only those references which are needed for the argumentation within the text. With the new references given under (5) I tried to achieve this, but I could not see that I have cited an outdated paper which needed to be updated.
All points have been additionally addressed within the new version of the Cover Letter!

Reviewer 2 Report
The paper “CoCoN-Database – a tool to assess nanomaterial hazard” by Krug constricts a new database to assess the potential hazard of nanomaterials or innovative materials and make the regulatory processes smoother and faster. The manuscript is well written and contains results that are represented by high quality. I recommended the publication of this manuscript after minor revision for the following reason.
- The abbreviation should be mentioned the complete first time
- The abstract should be concerned with actual results.
- The authors should be adding a clear hypothesis at the end of the introduction.
- In conclusion, the authors should explain how researchers benefit from this model and its applicability?
Author Response
- The abbreviation should be mentioned the complete first time
I explained not only the name of the database (new title) but also other abbreviations I have used within the text. - The abstract should be concerned with actual results.
see point 2 of the first reviewer; new abstract added - The authors should be adding a clear hypothesis at the end of the introduction.
I have adapted the last paragraph of the Introduction regarding this comment - In conclusion, the authors should explain how researchers benefit from this model and its applicability?
I have added essential parts within the “Conclusions” to address this point
I have addressed each point additionally within the new version of the Cover letter!

Round 2
Reviewer 1 Report
The author has carefully revised it according to the comments of the reviewer, and I agree to publish it after a small amount of language revision.
This manuscript is a resubmission of an earlier submission. The following is a list of the peer review reports and author responses from that submission.